# Risk of Malnutrition in Hospitalized COVID-19 Patients: A Systematic Review and Meta-Analysis

**DOI:** 10.3390/nu14245267

**Published:** 2022-12-10

**Authors:** Xiaoru Feng, Zeqi Liu, Xiaotong He, Xibiao Wang, Changzheng Yuan, Liyan Huang, Rui Song, You Wu

**Affiliations:** 1Institute for Hospital Management, Tsinghua University, Beijing 100084, China; 2School of Labor and Human Resources, Renmin University of China, Beijing 100872, China; 3Department of Occupational Hygiene Engineering, China University of Labor Relations, Beijing 100048, China; 4School of Public Health, Zhejiang University School of Medicine, Hangzhou 310058, China; 5Department of Nutrition, Harvard T. H. Chan School of Public Health, Boston, MA 02115, USA; 6School of Medicine, Tsinghua University, Beijing 100084, China

**Keywords:** risk of malnutrition, hospitalized COVID-19 patients, prevalence, meta-analysis

## Abstract

(1) Background: Studies have reported that COVID-19 may increase the risk of malnutrition among patients. However, the prevalence of such risk in hospitalized COVID-19 patients is uncertain due to the inconsistent use of assessment methods. (2) Methods: PubMed, Web of Science, and EMBASE were searched to identify studies on the nutritional status of hospitalized COVID-19 patients. A pooled prevalence of malnutrition risk evaluated by Nutrition Risk Score (NRS-2002) was obtained using a random effects model. Differences by study-level characteristics were examined by hospitalization setting, time of assessment, age, and country. Risk of bias was assessed using the Newcastle–Ottawa Scale. (3) Results: 53 studies from 17 countries were identified and summarized. A total of 17 studies using NRS-2002, including 3614 COVID-19 patients were included in the primary meta-analysis. The pooled prevalence of risk of malnutrition was significantly higher among ICU patients (92.2%, 95% CI: 85.9% to 96.8%) than among general ward patients (70.7%, 95% CI: 56.4% to 83.2%) (*p* = 0.002). No significant differences were found between age groups (≥65 vs. <65 years, *p* = 0.306) and countries (*p* = 0.893). (4) Conclusions: High risk of malnutrition is common and concerning in hospitalized patients with COVID-19, suggesting that malnutrition screening and nutritional support during hospitalization are needed.

## 1. Introduction

The coronavirus disease 2019 (COVID-19) pandemic, caused by the novel severe acute respiratory syndrome coronavirus 2 (SARS-CoV-2), has rapidly spread across the globe. The impact of COVID-19 is multi-dimensional, closely intertwined with the nutritional status both at the individual and population level. Changes in eating behavior and lifestyle, due to quarantine and social isolation, may lead to an impaired nutritional status [1]. Meanwhile, numerous studies have shown that malnutrition impacts viral replication and pathogenicity [2], induces atrophy of primary lymphoid organs, reducing T-cell and B-cell numbers, leading to leukopenia [3,4], and thus can increase the risk of illness following infection with SARS-CoV-2 [5]. Researchers around the world have reported a series of complications and sequelae of COVID-19, including dyspnea [6,7,8], cough [9], pulmonary fibrosis [10], pulmonary embolism [11], strokes [12,13], central nervous system (CNS) damage [14], cardiac arrhythmias [15], olfactory and taste disorders [16], and psychiatric disorders [17,18]. However, since the physiological risk caused by malnutrition is not immediate, and the assessment and monitoring of nutritional status are highly demanding on professional assessors, the increased risk of malnutrition during COVID-19 pandemic has not been paid enough attention.

The inpatient population is at high risk for malnutrition. It has been observed that nearly all hospitalized patients, including those with COVID-19, have poor nutritional status upon admission [19,20]. Malnutrition in patients is accompanied by low levels of lymphocytes, prealbumin, and albumin [20], which are significantly associated with a higher risk of transfer to intensive care unit (ICU) [21], a declined immune response and increased risk of infections [22], prolonged hospitalization [23], and poor prognosis [24]. Conversely, patients with critical symptoms of COVID-19 are at higher risk of malnutrition. Symptoms, including anorexia, diarrhea, vomiting, nausea, and mild abdominal pain, have been reported in COVID-19 patients [25], while antiviral drugs may exacerbate gastrointestinal symptoms [26]. These may result in an imbalance in energy intake and expenditure.

Though studies have addressed the potential mechanisms and associations between malnutrition and COVID-19 [23,27,28], nutritional assessment has not been generally incorporated in the clinical routine or the official guidelines of COVID-19 care [29,30,31]. Characterizing malnourished patients with COVID-19 is crucial to the management of the diagnosis, treatment, and to patient care in general, yet there is limited research on the prevalence of malnutrition or risk of malnutrition among patients with COVID-19, based on different population settings. Therefore, this systematic review and meta-analysis was conducted to estimate the pooled prevalence of malnutrition risk in hospitalized patients with COVID-19, including stratification by types of hospitalization, time of assessment, age, and country.

## 2. Methods

This systematic review was performed in accordance with Preferred Reporting Items for Systematic Reviews and Meta-Analysis (PRISMA) guidelines. The review protocol was registered with PROSPERO (CRD42022338383).

### 2.1. Search Strategy

We searched databases including PubMed, Web of Science, and EMBASE through 31 August 2022. To maximize the search for relevant articles, we also reviewed the reference lists of the relevant systematic reviews. The specific search strategy included combinations of the following MeSH and other key terms: “COVID-19”, “SARS-CoV-2”, “2019 Novel Coronavirus Disease”, “malnutrition”, “nutritional deficiency”, “nutritional status”. The search strategy is presented in Appendix A.

### 2.2. Eligibility Criteria

Studies were included in the initial systematic review if they met the following criteria: (1) population: adult (age ≥ 18 years) patients with a confirmed diagnosis of COVID-19; (2) exposure: COVID-19 infection leading to hospitalization; (3) outcome: malnutrition or risk of malnutrition identified by validated assessment tools; and (4) study type: cohort studies and cross-sectional studies. In the full text review, studies were excluded if they did not report the prevalence of malnutrition or risk of malnutrition.

### 2.3. Study Selection

The two independent reviewers (X. F. and X. H.) first screened the titles and abstracts to identify potentially relevant studies. In the next step, the full texts were evaluated according to the inclusion and exclusion criteria. Disagreements were resolved by discussion and consensus with a third reviewer (Y. W.).

### 2.4. Quality Assessment

All included studies were checked for risk of bias by two independent reviewers (Z. L. and X. W.). In case of disagreements, a third reviewer was consulted for assistance (Y. W.). We used the Newcastle–Ottawa Scale for assessing the quality of studies in meta-analysis, with the evaluation dimensions adapted to cross-sectional studies and to our research question. The dimensions included selection (representativeness, description of non-respondents/excluded patients, and ascertainment of the exposure); comparability (control for important and additional factors); and outcome (the use and data collection of the assessment of tool, and the statistical parameter). Each dimension was assigned one point; a maximum of eight points could be granted if a study satisfied all requirements. In the sensitivity analysis, the results of all studies and the studies with ≥6 points were analyzed and compared.

### 2.5. Data Extraction

Two independent researchers (Y. W. and X. F.) extracted the following data from the included studies, and disagreements were resolved by consensus: (1) article information: the last name of the study’s first author and publication year; (2) study information: country, study design(cohort study or cross-sectional study), and time frame; (3) population information: age, sex, race/ethnicity, population size, population source (general ward patients or ICU patients), and severity of COVID-19; and (4) malnutrition information: timing of assessment (at admission or during hospitalization), assessment tool, and prevalence data. When there were both general ward patients and ICU patients in a study, they were considered separately.

### 2.6. Statistical Analysis

Since heterogeneity in malnutrition prevalence was expected across populations, we performed a random effects model to estimate the pooled prevalence of risk of malnutrition and the 95% confidence intervals. The prevalence data were pooled separately by each assessment method. To detect any difference between the settings of hospitalization and the timings of nutritional status assessment with enough power, we quantitatively synthesized studies using NRS-2002, as it was the most commonly used tool. NRS-2002 score ≥ 3 is considered at risk of malnutrition (NRS 3–4: moderate risk of malnutrition; NRS ≥ 5: severe risk of malnutrition). All analyses were performed for general ward patients and ICU patients separately.

We assessed between-study heterogeneity using Q statistic and the I2-test statistic. Furthermore, to identify additional possible sources of heterogeneity, subgroup analyses were performed according to the World Bank country classification by income level and the median or mean age (≥65 and <65 years) of the study population, respectively. Heterogeneity between groups were assessed by chi-squared test [32].

For meta-analysis of proportion data, studies have shown that the visual appearance of funnel plots and the sensitivity of Egger’s regression to detect asymmetry are prone to misclassification [33,34]. A Doi plot, instead, was proposed to find out any publication bias in pooled analysis of proportions, and the risk can be further quantified using the Luis Furuya-Kanamori (LFK) index [35]. An LFK index within ±1, out of ±1 but within ±2, and out of ±2 indicates no asymmetry, minor asymmetry, and major asymmetry, respectively.

All data were analyzed using Stata 15.0 (Stata Corporation, College Station, TX, USA) and MetaXL 5.3 (EpiGear International, Noosa, Queensland, Australia). We used the metaprop command in Stata as it allows inclusion of studies with proportions equal to zero or 100 percent and avoids confidence intervals beyond the 0 to 1 range. To stabilize the variance, all analyses adopted the Freeman–Tukey double arcsine transformation [32]. For all hypothesis tests, *p* < 0.05 was considered statistically significant.

## 3. Results

### 3.1. Search Results

The systematic literature search identified a total of 4429 articles in the three databases. After removing 954 duplicates, the titles and abstracts of 3475 articles were screened, of which 2588 were excluded due to inappropriate subject matter. The remaining publications were examined based on the inclusion and exclusion criteria. A total of 53 articles using various assessment tools (Body Mass Index [BMI], Controlling Nutritional Status Score [CONUT], Global Leadership Initiative on Malnutrition [GLIM], Geriatric Nutritional Risk Index [GNRI], Mini Nutritional Assessment [MNA], Modified Nutrition Risk in the Critically ill [mNUTRIC], Malnutrition Universal Screening Tool [MUST], Nutrition Risk Screening-2002 [NRS-2002], and Subjective Global Assessment [SGA]) are summarized in Appendix A. Significant differences between the measurement tools (*p* < 0.001, Appendix A) suggested that the prevalence measured by the above-mentioned instruments were not comparable. A total of 17 articles using NRS-2002 were included in the quantitative meta-analysis to compare the prevalence of malnutrition risk assessed at different time hospitalization settings and points. The flow diagram for the literature review and article selection process following the PRISMA guideline is shown in Figure 1.

### 3.2. Study Characteristics

Appendix A summarizes the 53 studies of interest by all assessment tool. Table 1 presents the characteristics of the studies using NRS-2002 [36,37,38,39,40,41,42,43,44,45,46,47,48,49,50,51,52]. These 17 studies included in the primary meta-analysis were from nine different countries. A total of 3614 COVID-19 patients were included in these studies with median/mean age ranging from 45.7 to 86.1 years. Regarding population sources, 13 studies recruited patients in the general wards and six studies recruited patients in ICUs.

### 3.3. Quality Assessment

The results of the quality assessment are shown in Figure 2. 15 out of 17 studies are ≥6 points. Most of the individual studies lost points because the protocols of data collection in the outcome assessment were not detailed (nine out of 17 studies). When analysis was limited to studies with six points and above, the pooled prevalence changed by less than 10% across groups.

### 3.4. Prevalence of Risk of Malnutrition among Hospitalized Patients with COVID-19

#### 3.4.1. Overall Results

The pooled prevalence of malnutrition risk among patients with COVID-19 as assessed by NRS-2002 is shown in Figure 3. The pooled prevalence of ICU patients (92.2%, 95% CI: 85.9% to 96.8%) was significantly higher than that of general ward patients (70.7%, 95% CI: 56.4% to 83.2%) (*p* = 0.002). For general ward patients, there were not significant between-group differences for subgroup analysis by time of assessment for prevalence of risk of malnutrition (*p* = 0.083, Appendix A). Pooled prevalence of malnutrition or risk of malnutrition as assessed by other instruments ranged from 11.8–83.3% for general ward patients and 31.5–94.4% for ICU patients (Appendix A).

#### 3.4.2. Subgroup Analyses

To identify the prevalence of malnutrition risk in COVID-19 patients of different ages, 12 studies were grouped by median/mean age ≥65 vs. <65 years. Studies with median/mean age ≥ 65 years (80.5%, 95% CI: 64.0% to 92.9%) showed higher pooled prevalence of malnutrition risk than those with median/mean age <65 years (64.1%, 95% CI: 34.6 to 88.8%), although this was not statistically significant (*p* = 0.306) (Figure 4).

According to the World Bank criteria for classifying countries by income level, the individual studies were stratified into three categories: high-income economies (Switzerland, Spain, Italy, and Poland), upper-middle-income economies (China and Brazil), and lower-middle-income economies (Philippines, Egypt, and Iran). The pooled prevalence in the high-income economies, upper-middle-income economies, and lower-middle-income economies was 78.9% (95% CI: 65.0% to 90.0%), 81.5% (95% CI: 70.3% to 90.5%), and 73.6% (95% CI: 35.0% to 98.2%), respectively, indicating little difference between countries by income level (*p* = 0.893) (Figure 5).

### 3.5. Publication Bias

Risk of bias assessment for all studies and two subgroups were visualized by Doi plots (Appendix A). Only minor asymmetry was found for general ward patients (LFK = −1.91), whereas no evidence of publication bias was found in the case of all studies (LFK = −0.83) and the studies of ICU patients (LFK = 0.97).

## 4. Discussion

Hospitalized patients are a high-risk group for malnutrition [53]. Malnutrition can result in some adverse consequences, including hypercatabolism, rapid muscle wasting [54], weakened immune response [54], longer hospital length of stay [55], and higher hospital mortality [56]. The COVID-19 pandemic has brought about a spike in hospitalizations, calling for a closer monitoring of nutritional status in patient care.

### 4.1. Main Findings

In our study, we comprehensively summarized the prevalence of malnutrition and risk of malnutrition assessed by different tools and conducted a quantitative meta-analysis of the risk of malnutrition measured by NRS-2002, including 17 studies from nine countries with 3614 hospitalized COVID-19 patients from general wards and ICU. The pooled prevalence of malnutrition risk of general ward patients and ICU patients was 70.7% and 92.2%, respectively.

Our results are in line with other studies that suggested higher prevalence of malnutrition risk among hospitalized COVID-19 patients compared with hospitalized patients of other diseases [53,57]. Unlike previous qualitative reviews [24,53,58] on the nutritional status of COVID-19 patients, we conducted a quantitative meta-analysis to estimate the pooled prevalence of risk of malnutrition.

Risk of malnutrition was found more prevalent among ICU hospitalized COVID-19 patients than general ward patients, suggesting that malnutrition might be of a greater threat to COVID-19 patients with severe symptoms. Therefore, regular assessment of the nutritional status in hospitalized patients, especially in ICU patients, is highly recommended. We did not observe significant difference by income or age group, suggesting that malnutrition is a common phenomenon that should be treated equally across age groups and countries. Although the age subgroup analysis was not statistically significant, since the elderly population is inherently at higher risk of diabetes, low albumin, and vitamin D deficiency [28,59], the consequences of malnutrition risk could be more severe for them. We recommend that more attention be paid to this population.

### 4.2. Assessment Tools for Malnutrition and Risk of Malnutrition

Nutritional screening measures used in COVID-19 patients can be divided into three categories: (1) Traditional nutritional screening measures: NRS-2002, MNA, GLIM, mNUTRIC, and MUST; these tools have full nutritional assessment, and are applied by specialized professionals, such as nutritionists and doctors; (2) Calculated nutritional indices: PNI and COUNUT; they are based on one routine investigations and easy to calculate without a need for sophisticated skills; (3) Combinations of clinical characteristics, anthropometric measures, and nutritional biomarkers [60]. Among the most widely used tools, NRS-2002 score is a composition of nutritional score, disease severity score, and age-adjusted score, suitable for assessing the risk of malnutrition [61]; for rapid screening of hospitalized patients, NRS-2002 is considered a relatively mature instrument that has better feasibility and predictive validity for prolonged hospitalization [53]. For formal diagnosis of malnutrition, a more comprehensive GLIM evaluation is recommended [21,55].

In recent years, in order to obtain the nutritional status of hospitalized patients in real time, some information technologies have been used for nutritional monitoring of hospitalized patients. One study developed a tablet app as a tool to monitor dietary intake in hospitalized patients at nutritional risk [62]. Another study used a new clinical decision support system that combined collected clinical data with patient-generated data from a smartphone app to monitor patients’ nutritional status [63]. These digital tools and applications can reduce the workload and time spent on nutritional assessments for healthcare professionals.

### 4.3. Recommendations following Malnutrition Screening

In addition to early assessment of nutritional status and identification of nutritional risks through various screening tools, the nutritional status of COVID-19 patients can also be optimized by conducting nutritional counseling with professionals such as registered dietitians and clinical dietitians [64]. According to European Society for Clinical Nutrition and Metabolism (ESPEN) guidelines, COVID-19 patients diagnosed with malnutrition should receive adequate vitamin and mineral supplementation, including vitamins A, B, C, D, omega-3 polyunsaturated fatty acids and micronutrients such as selenium, zinc and iron [64]. Patients diagnosed with moderate malnutrition should take oral nutritional supplements (ONS) in addition to dietary multivitamins and micronutrients. For severely malnourished patients, enteral nutrition (EN) or ONS should be provided if respiratory status contraindicates EN [53]. Besides, regular physical activities are encouraged to maintain physical and mental health as well as muscle mass and body composition.

For general ward inpatients, ESPEN encourages oral feeding rather than EN or parenteral nutrition (PN) unless contraindicated [64]. A variety of high-calorie diets as well as easily digestible foods and snacks can be offered to COVID-19 patients to increase protein-calorie intake, such as yogurt, mousse, cheese, etc. For patients with eating difficulties, ONS and intravenous multivitamins, multiminerals, and trace element solutions are options available [65].

For patients in ICU, although oral is the preferred feeding route, EN or PN are recommended immediately after ICU admission when contraindications are present. ESPEN recommends feeding within 48 h of ICU admission for COVID-19 ICU patients [64], while American Society for Parenteral and Enteral Nutrition (ASPEN)/Society of Critical Care Medicine (SCCM) believes EN should be provided within 24–36 h of ICU admission or within 12 h of intubation and placement on mechanical ventilation [66]. Because of the complex nutritional status of critically ill patients with COVID-19, all patients may require individualized diet-led nutritional interventions during their ICU stay [67].

Our study found that older populations tended to have a higher risk of malnutrition, although not significantly. Elderly COVID-19 patients are prone to high nutritional risk due to higher prevalence of comorbidities, aging-related changes in body composition, and decreased muscle mass [68]. ESPEN guidance for elderly patients with COVID-19 recommends that nutrition therapy should be started as early as possible after admission (within 24–48 h). For elderly patients whose nutritional status may have been compromised, nutritional therapy should be introduced gradually to prevent refeeding syndrome. In addition, evidence suggested that oral nutritional supplements (ONS) might have a slightly positive effect on energy intake, protein intake and mobility [69], and EN should be administered while potential complications is being monitored. Also, PN should be considered when EN fails to achieve the target [64].

To our knowledge, this is the first study to quantify the synthesize prevalence of malnutrition risk in COVID-19 patients. Based on our inclusion/exclusion criteria, all studies included (1) have good representativeness of the adult COVID-19 patient population; (2) used validated instruments for nutritional assessment (NRS-2002); and (3) presented detailed subgroup prevalence of risk of malnutrition in hospitalized patients with COVID-19. This allowed us to analyze subsets of the studies that have presented subgroup data for different stratification factors. All of these made our results more reliable and credible.

Our study has limitations as well. In quality assessment, we found that a few studies had relatively unsatisfactory reporting of the specific use of the NRS-2002 instrument and the details of the non-respondents. However, the sensitivity analysis including only the high-quality studies suggested little influence by such concerns. We recommend that future studies provide the protocol of data collection and any adaptation of the validated tools to the study population. Meanwhile, future studies may report number/proportion of patients excluded due to particular reasons, so that the representativeness of the study population could be justified. In the clinical settings of the included studies, the nutritional status of all patients before COVID-19 diagnosis was unknown. Therefore, the risk of malnutrition could not be considered as a consequence of COVID-19, but rather an observable variable that requires more attention, especially for those hospitalized in ICU. Although NRS-2002 is the most commonly used tool to assess risk of malnutrition, it was not routinely done in several major countries such as US, Australia, and Canada, where MUST, MST, mNUTRIC were used more often. Finally, we were not able to analyze the differences between different COVID-19 waves due to restriction by the time frame and the information available in the included studies. In future research, a comparative analysis on different COVID-19 waves, such as Alpha, Beta, Gamma, Delta, and Omicron periods would add new insights to the research topic.

## 5. Conclusions

Our results indicated that high risk of malnutrition is a common and concerning phenomenon in hospitalized patients with COVID-19, which highlights the importance of monitoring malnutrition and providing nutritional support during hospitalization. Risk of malnutrition is prevalent in all hospitalized COVID-19 patients regardless of the countries of origin, and the elderly remain a group deserving particular attention in inpatient care.

## Figures and Tables

**Figure 1 nutrients-14-05267-f001:**
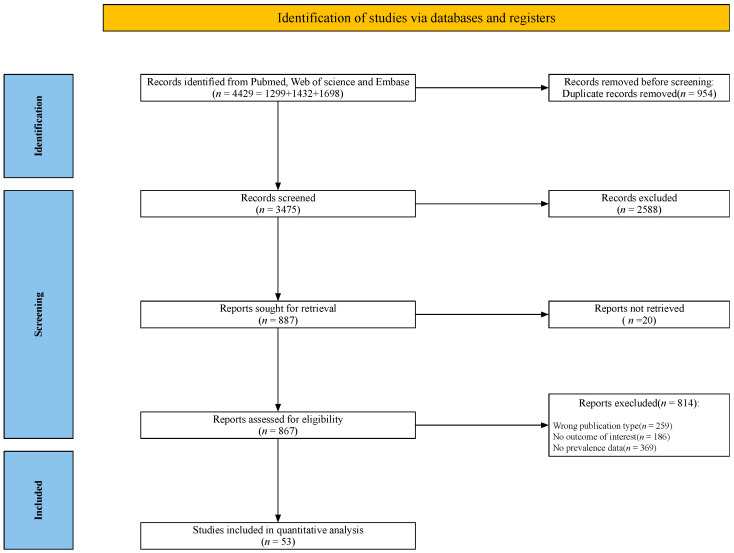
Flow chart of literature review and study selection.

**Figure 2 nutrients-14-05267-f002:**
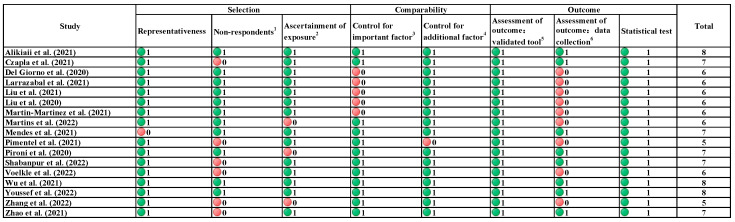
Quality assessment by Newcastle–Ottawa Scales (Green indicates score, red indicates no score) [36,37,38,39,40,41,42,43,44,45,46,47,48,49,50,51,52]. ^1^ Non-respondents: one point is granted if the individual study included data from all patients during the study period, or if they specified the number/proportion of patients excluded from the analysis. ^2^ Ascertainment of exposure: one point is granted if COVID-19 diagnosis/ascertainment method is clearly described in the individual study. ^3^ Control for important factor: In this meta-analysis, the primary stratification factor is the source of patients (ICU/general ward). One point is granted if the individual study specified the patient source, or when the study population included patients from mixed sources, specified the prevalence of malnutrition in each subgroup. ^4^ Control for additional factor: one point is granted if the individual study further presented subgroup analyses by sex, age groups, COVID-19 severity, etc. ^5^ Assessment of outcome (validated tool): All studies included used NRS-2002 to evaluate the nutritional status of the patients, therefore one point is granted for all. ^6^ Assessment of outcome (data collection): one point is granted if the individual study clearly described the protocol of NRS-2002 evaluation, include but not limited to the measurement of height, weight, and food intake by experienced nutritionists.

**Figure 3 nutrients-14-05267-f003:**
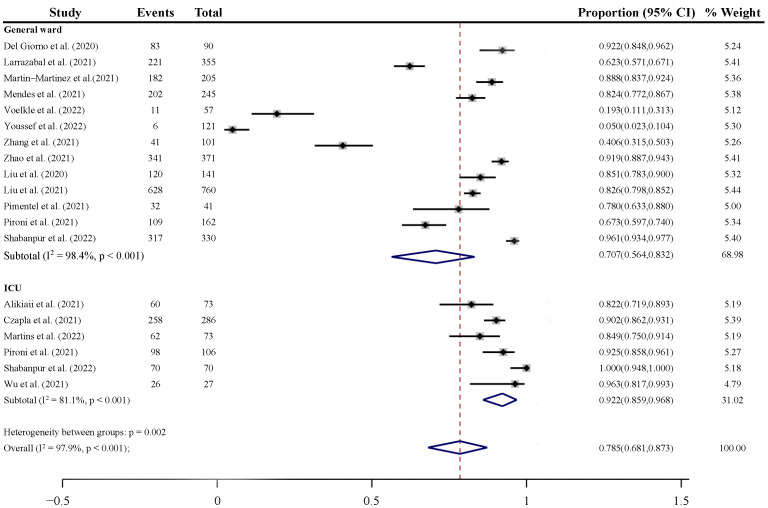
The pooled prevalence of malnutrition in COVID-19 patients [36,37,38,39,40,41,42,43,44,45,46,47,48,49,50,51,52].

**Figure 4 nutrients-14-05267-f004:**
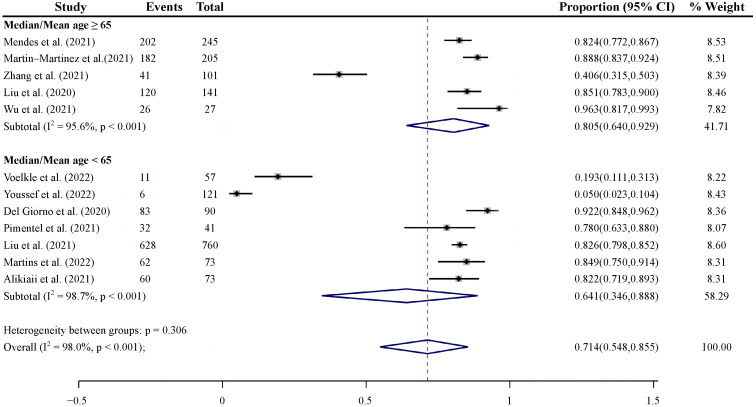
Subgroup analysis of the pooled prevalence of malnutrition in COVID-19 patients by medium/mean age [36,38,39,40,41,43,44,45,46,49,51,52].

**Figure 5 nutrients-14-05267-f005:**
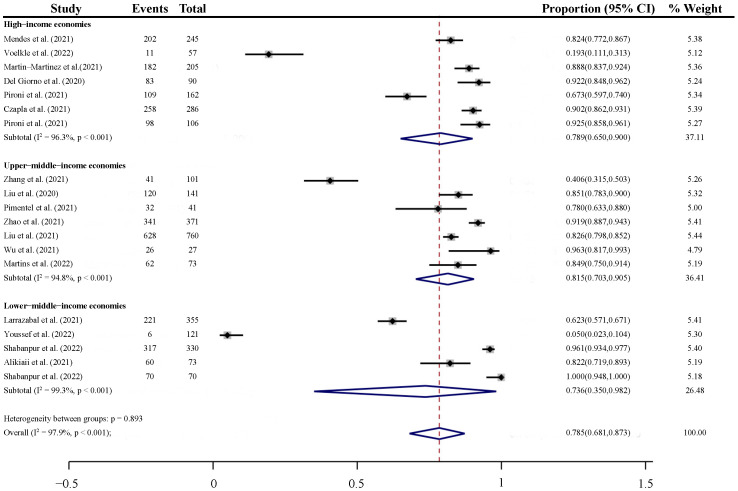
Subgroup Analysis of the Prevalence of Malnutrition in COVID-19 Patients by World Bank country classifications by income level [36,37,38,39,40,41,42,43,44,45,46,47,48,49,50,51,52]. High-income economies: Switzerland, Spain, Italy, and Poland; Upper-middle-income economies: China and Brazil; Lower-middle-income economies: Philippines, Egypt, and Iran.

**Table 1 nutrients-14-05267-t001:** Study characteristics of hospitalized COVID-19 patients assessed by Nutritional Risk Score-2002.

Author	Country	Time Frame	Sample Size	Age ^1^	Sex	COVID-19 Confirmation	Timing of Assessment	Specific Time of Evaluation	Outcome ^2^	Prevalence ^3^
**General Ward**										
Del Giorno et al. (2020) [36]	Switzerland	2020/3	90	64.5 ± 13.7	Male (67.8%)	PCR ^4^/CT ^5^	At admission	the first 24 h of admission	Malnutrition risk (Y/N)	92.0%
Larrazabal et al. (2021) [37]	Philippines	2020/7/15–2020/9/15	355	NR	NR	RT-PCR ^6^	At admission	NR ^7^	Malnutrition risk (low/moderate/high)	37.7% low; 47.3% moderate; 14.9% high risk
Martin–Martinez et al. (2021) [38]	Spain	2020/4/14–2020/7/30	205	69.3 ± 17.5	Male (47.8%)	PCR	At admission	NR	Malnutrition risk (Y/N)	88.7%
Mendes et al. (2021) [39]	Switzerland	2020/3/13–2020/5/17	245	86.1 ± 6.4	Male (42%)	RT-PCR	At admission	NR	Malnutrition risk (no/at risk/high risk)	17.6% no risk; 32.2% at risk; 50.2% high risk
Voelkle et al. (2022) [40]	Switzerland	2020/3/17–2020/4/30	57	67.0 (60.0, 74.2)	Male (60%)	RT-PCR	At admission	NR	Malnutrition risk (Y/N)	19.0%
Youssef et al. (2022) [41]	Egypt	2020/7–2020/12	121	52.4 ± 10.5	Male (84.3%)	NR	At admission	on day 1 of admission	Malnutrition risk (Mild–moderate/Severe)	94.9% Mild–moderate; 5.1% Severe
Zhao et al. (2021) [42]	China	2020/1/29–2020/2/19	371	NR	NR	NR	At admission	on the first day of hospitalization	Malnutrition risk (no/low/high)	8% no risk; 76% low; 16% high risk
Zhang et al. (2021) [43]	China	2020/2/6-2020/3/20	101	65.3 ± 13	Male (59.4%)	NR	At admission	on the first day of hospitalization	Malnutrition risk(Y/N)	40.5%
Liu et al. (2020) [44]	China	2020/1/28–2020/3/28	141	71.7 ± 5.9	Male (48.2%)	RT-PCR/CT	During hospitaliza-tion	NR	Malnutrition risk (Y/N)	85.1%
Liu et al. (2021) [45]	China	2020/1/29–2020/3/15	760	60 (46, 74)	Male (50%)	RT-PCR/CT	During hospitaliza-tion	NR	Malnutrition risk (Y/N)	82.6%
Pimentel et al. (2021) [46]	Brazil	NR	41	45.7 ± 12.4	Male (61.0%)	RT-PCR	During hospitaliza-tion	NR	Malnutrition risk (low/high)	22.0% low risk; 78.0% high risk
Pironi et al. (2021) [47]	Italy	2020/4	162	NR	NR	NR	During hospitaliza-tion	a one-day clinical audit	Malnutrition risk (Y/N)	67.3%
Shabanpur et al. (2022) [48]	Iran	2021/5–2021/7	330	NR	NR	NR	During hospitaliza-tion	in a 6-week period	Malnutrition (risk/moderate/severe)	4% at risk; 69% moderate; 27% severe risk
**ICU**										
Alikiaii et al. (2021) [49]	Iran	2021/1/1	73	58.9 ± 18.8	Male (63%)	RT-PCR	During hospitaliza-tion	NR	Malnutrition risk (low/moderate/high)	17.8% low; 69.9% moderate; 12.3% high risk
Czapla et al. (2021) [50]	Poland	2020/9–2021/6	286	NR	Male (67.8%)	RT-PCR	At admission	at the time of admission to ICU ^8^	Malnutrition risk(Y/N)	90.2%
Martins et al. (2022) [51]	Brazil	2020/3–2020/10	73	56	Male (63%)	PCR	At admission	the first 48 h of admission to the ICU	Malnutrition risk(Y/N)	85.0%
Pironi et al. (2021) [47]	Italy	2020/4	106	NR	NR	NR	During hospitaliza-tion	a one-day clinical audit	Malnutrition risk(Y/N)	92.5%
Shabanpur et al. (2022) [48]	Iran	2021/5–2021/7	70	NR	NR	NR	During hospitaliza-tion	in a 6-week period	Malnutrition (risk/moderate/severe)	0% at risk; 20% moderate; 80% severe risk
Wu et al. (2021) [52]	China	2020/1/15–2020/2/29	27	74.9 ± 10.5	Male (66.7%)	NR	During hospitaliza-tion	on the seventh day of admission	Malnutrition risk (potential/ high)	3.7% potential risk; 96.3% high risk

^1^ Data are presented as mean ± SD or median (IQR) unless otherwise specified. ^2^ NRS-2002 score ≥ 3 is considered at risk of malnutrition (NRS 3–4: moderate risk of malnutrition; NRS ≥ 5: severe risk of malnutrition). ^3^ Prevalence was extracted from the original results of each publication. In subsequent analyses, unified prevalence of risk of malnutrition was calculated as the proportion of patients with NRS-2002 score ≥ 3. ^4^ PCR, polymerase chain reaction. ^5^ CT, computed tomography. ^6^ RT-PCR, reverse transcription polymerase chain reaction. ^7^ NR, not reported. ^8^ ICU, intensive care unit.

## Data Availability

Data used for the analysis are available from the corresponding authors on request.

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
