# Peer review of "Risk of Malnutrition in Hospitalized COVID-19 Patients: A Systematic Review and Meta-Analysis"

_nutrients, 2022, doi:10.3390/nu14245267_

Round 1
Reviewer 1 Report
Nutrients Article Review
Thank you for patience in slight delay in submitting review for this meta-analysis titled Risk of Malnutrition in Hospitalized COVID-19 Patients: A Systematic Review and Meta-analysis.
The methodology to identify and evaluate available literature based on stated criteria is valid
The analysis methodology is valid.
Suggestions are provided for consideration prior to publication.
Title revision (item 1) and limitations (items 3-5) should be required for clarity prior to publication.
Suggestions for improvement:
1. Title revision suggestion Risk of Malnutrition Assessed by NRS-2022 in Hospitalized COVID-19 Patients: A Systematic Review and Meta-analysis.
2. Table 1: define how prevalence was calculated in the legend
3. Clarify how “malnutrition” was determined; Table 1 infers y and no based on NRS-2002. Authors need to expand text to describe the NRS-2002 rating system or at a minimum in footnote for Table.
4. Lines 282-284. Malnutrition status as time of admission is unknown. Authors need to add this to the limitations. Although countries were sorted by criteria to help with economic bias, the authors should acknowledge potential confounding variable at time of admission. No data has been provided in the analysis to confirm malnutrition was a result or linked to COVID-19, rather was present as an observable variable.
5. Line 372+ Limitations should also acknowledge the issue with lack of data from three major countries: Australia, Canada, and the US. Was this because NRS-2002 was not used as the malnutrition assessment or? This is a major limitation of the analysis and should be added. This is not the fault of the authors, it appears, but of the global NRS-2002 use.
6. Line 184+ Authors might consider removing the names of the selected countries included in the analysis. Refer to Table 1 where countries are listed.
Reviewer 2 Report
Dear authors!
The difference of a good article lies in the precise task, adequate research methods and, as a result, correct conclusions. Your article makes a great contribution to solving the problem of treating difficult patients with Covid 19. Thank you very much!
Author Response
We are really grateful that the reviewer took the time to read our manuscript and showed positive evaluation of the work. As suggested by reviewers, we have provided more details to improve our study. We hope that the 2nd version will be more qualified than the original manuscript to be accepted by the journal.